# Effect of Citric Acid Cross Linking on the Mechanical, Rheological and Barrier Properties of Chitosan

**DOI:** 10.3390/molecules27165118

**Published:** 2022-08-11

**Authors:** Nusrat Sharmin, Jan Thomas Rosnes, Leena Prabhu, Ulrike Böcker, Morten Sivertsvik

**Affiliations:** 1Department of Food Safety and Quality, Nofima AS, Osloveien 1, 1430 Ås, Norway; 2Department of Processing Technology, Nofima AS, Richard Johnsens gate 4, 4021 Stavanger, Norway; 3Department of Raw Materials and Process Optimisation, Nofima AS, Osloveien 1, 1430 Ås, Norway

**Keywords:** chitosan, citric acid, acetic acid, cross-linking

## Abstract

In this study, acetic acid (AA-2% *w*/*v*), a combination of acetic acid and citric acid (AA-1% *w*/*v* + CA-1% *w*/*w*), and three different concentrations of citric acid (CA-2, 4 and 6% *w*/*w*) were used to create chitosan solution. The FTIR analysis showed the presence of residual CA in all the CA-containing samples where no trace of AA was observed. The tensile strengths of the CA-containing samples were lower than the AA samples. Whereas the values for the elongation at break of the CA samples were higher than the AA samples, which kept increasing with an increasing CA content due to the plasticizing effect from residual citric acid. The elongation at break values for 4 and 6% CA-containing samples were 98% higher than the AA samples. The samples prepared with CA showed shorter LVE regions that reduced with an increasing CA concentration compared to the AA samples. Different acid concentrations did not have a large effect on the gelation time. However, CA-containing samples showed higher viscosities as compared to the AA-containing solution, which increased with an increasing CA content. The water vapour transmission rates of the CA-containing samples were lower than the others. All the chitosan solutions suppressed the growth of the two test strains, and none of the variants reached an abs 600 nm at 0.2.

## 1. Introduction

In recent years, the concern for environmental issues and the challenges related to petroleum-based traditional plastic materials have triggered the development of biodegradable films from renewable sources that are readily available in nature. The most studied bio-based polymers include polysaccharides, proteins, and lipids [1]. Among them, the polysaccharides are of the most interest due to their low-cost, abundance, good film-forming ability, and relatively stable properties [2]. Chitosan is a nontoxic, bifunctional, biocompatible, and biodegradable pH-dependent cationic amino polysaccharide [3]. Chitosan, obtained via the alkaline deacetylation of chitin, is the second most abundant biopolymer on earth after cellulose. The structure of chitosan is composed of randomly distributed D-glucosamine and *N*-acetyl-D-glucosamine units linked by β-(1 → 4) glycosidic bonds [4]. This polysaccharide has attracted growing amounts of attention in the agricultural, food, industrial, and medicinal fields as a functional substance [4,5]. Chitosan has inherent antimicrobial properties against a wide range of bacteria, good gel and film-forming properties, good mechanical properties, and selective permeability in water and gases, which makes it a very lucrative option for food packaging applications [6,7,8].

Chitosan is a weak base and is not soluble in water or organic solvents. An acidic aqueous media (pH < 6) is required to solubilize chitosan. Above pH 6 chitosan is insoluble due to the deprotonated amines. The acidic media protonates the amino groups present in chitosan’s structure, making the polymer positively charged and thereby weakening the associated forces between chitosan chains [9]. After chitosan is completely dissolved in the acidic aqueous solution, it can form weak intermolecular links through hydrophobic interactions between residual acetyl groups [9]. The chain flexibility of chitosan is sensitive to the type of counterion (i.e., the acid ions) [10]. Therefore, the type of solvent used to dissolve chitosan has an important influence on the rheological properties of chitosan solutions [11,12]. The solvent’s use can also alter the properties of the polymeric films [13]. Although diluted acetic acid is mostly used to make chitosan solution, several other organic acids including lactic, formic, L-glutamic, and malic acid have been reported to effectively dissolve chitosan [14].

Crosslinking is a promising and highly recommended technique for improving the performance and applicability of polysaccharide-based films. The mechanical and barrier properties of chitosan can be improved via crosslinking with different cross-linking agents. In addition, the addition of plasticizers can improve the flexibility and chain mobility of the chitosan films by decreasing the intermolecular forces along the polymer chain. The most common cross-linking agents used for chitosan are glutaraldehyde and genipin [15]. Glutaraldehyde can increase the functional properties of chitosan through a crosslinking reaction between aldehyde and amino groups [16]. However, such cross linking can hamper the biocompatibility of chitosan. On the other hand, crosslinking with genipin can enhance its mechanical strength, surface hydrophilicity, and hydration ability, and such cross linking does not affect the biocompatibility of chitosan [17]. However, the use of genipin as a crosslinking agent for chitosan is very limited due to its high cost. Citric acid (CA) is a readily available biobased polycarboxylic acid extracted from citrus fruits, which shows better biocompatibility as compared to other cross-linking agents [18].

In recent years, attempts have been made to use citric acid to cross link polysaccharides in order to improve their mechanical properties. Citric acid is a non-toxic weak tricarboxylic acid, which is organic in nature, and is approved by the Food and Drug Administration [19]. Citric acid is a natural acid with the ability to cross-link and stabilize polysaccharide structures by creating covalent intermolecular di-ester linkages with the hydroxyl groups of the polysaccharide [20]. Such kinds of cross-linking can increase the elasticity and mechanical strength of chitosan films. Wen et al. studied the effect of citric acid cross linking on the functional properties of multifunctional food packaging films based on carboxymethyl chitosan and polyvinyl alcohol [21]. They reported that the addition of citric acid not only increased the mechanical and antimicrobial properties of the developed films but also helped to promote the soil-related microbial degradation of the composite films, which showcases their potential application value as a food packaging material. Yao et al reported that the addition of citric acid at a low concentration can improve the water solubility, water contact angle, mechanical properties, water vapor permeability, oxygen barrier properties, and heat sealing characteristics of mung bean-starch edible films [22].

Most studies have reported the dissolution and cross linking of chitosan to be two-step processes. In the first step, the chitosan is dissolved in a diluted organic acid solution, and in the second step the crosslinking agent is added to the solution to achieve improved functional properties, which makes it a time-consuming method. The main aim of this study was to establish a one step process of dissolving and crosslinking chitosan using a combination of acetic and citric acid and different concentrations of citric acid. Wei et al. also developed a one-step strategy to prepare a chitosan-based hydrogel in which citric acid was used simultaneously as a dissolving and crosslinking agent [23]. However, they studied the effect of the addition of citric acid on the physiochemical properties of chitosan with different molecular weights. They did not explore the effect of different citric acid concentrations or a combination of acetic acid and citric acid on chitosan properties.

It is well known that the polymeric films should have good mechanical and barrier properties in order to qualified for use in food packaging applications. The rheological properties of the polymer blends also play a key role in deciding the applicability of a particular polymer solution for food packaging applications, particularly for their use in edible coatings. The secondary objective of our current study was to observe how such a one step process can alter the functional properties of chitosan. Different concentrations of citric acid were used to dissolve and cross link chitosan. The effects of different citric acid concentrations on the structural, mechanical, rheological, barrier-related, and antimicrobial properties of chitosan were observed. In addition, chitosan solution was also prepared using acetic acid. The functional properties of chitosan prepared using acetic acid were also evaluated and compared with the citric acid samples.

## 2. Materials and Methods

### 2.1. Materials

Chitosan (Deacetylated chitin and Poly-D-glucosamine) with medium molecular weight (75–85% deacetylated; molecular weight—190–310; viscosity—200–800cP) and citric acid (99%; molecular weight—192.12) were purchased from Sigma-Aldrich, Norway. Acetic acid (glacial) 100% was purchased from Merck, Germany. DPPH (2, 2-Diphenyl-1-picrohydrazyl) was purchased from Alfa Aesar, Germany.

### 2.2. Film Preparation

The film-forming solution was prepared by adding 1 g chitosan powder into the acidic solution prepared using acetic acid, a combination of acetic acid and citric acid, or citric acid. The solution was then heated to 70 °C with constant stirring at 800 rpm for 1 h. The pH of the solution was measured using a FiveEasy Plus pH meter (Mettler Toledo, Columbus, OH, USA) equipped with a LE410 electrode. A total of 20 mL of the resulting solution was then poured into a 90 mm polystyrene Petri dish and left to dry under a fume cupboard for 48 h. Table 1 provides the sample codes and corresponding pH used throughout the study.

### 2.3. FTIR Analysis

FTIR measurements were carried out on a Bruker INVENIO^®^ Spectrometer (Bruker Optics, Germany) equipped with a high-throughput extension HTS-XT. Samples were measured as thin films on 96-well Si plates with 5 replicate measurements for each sample. Data acquisition was controlled by OPUS v8.5 software. Spectra were collected in transmission mode in the range 4000–600 cm^−1^ with a resolution of 4 cm^−1^. Before each sample measurement, a background spectrum of the empty Si plate was collected; the number of scans for both background and sample was 40. In order to achieve optimal signal response, a sample volume of 6 μL per well for CH-2% AA and 2 μL for CH-1%AA-1%CA was used. For CH-2%CA, CH-4%CA, and CH-6%CA the volume was 2 μL after diluting with H_2_O 1:1 to avoid detector saturation. Baseline correction and averaging of the five replicate spectra were performed with UnscramblerTM v11 (Camo Analytics AS, Oslo, Norway) software.

### 2.4. Mechanical Properties

The mechanical properties of the films (tensile strength and elongation at break) were conducted using a TA.XT plus texture analyzer (Stable Micro Systems Ltd., Godalming, UK) following the ASTM D638-699 method (1999). The texture analyzer was equipped with a 50 kg load cell and a crosshead speed of 1 mm/s and span distance of 25 mm was used for the experiment. Triplicates of each sample were performed and analyzed using the software Exponent ver: 6.1, 16.0. The thickness of the films was measured with a Japan Mitutoyo 500-197-20/30 200 mm/ 8″ Digital Digimatic Vernier Caliper (0.01 mm resolution; ±0.02 mm accuracy).

### 2.5. Rheological Properties

A hybrid rheometer (Discovery HR-2, TA Instruments, Newcastle, UK) and cone and plate geometry (40 mm, 2°) was used for the analysis of the rheological properties. The experiment was conducted at 22 °C. Roughly 1 mL test sample was loaded onto the cross-hatched Peltier plate and subjected to frequency and amplitude sweeps. The ranges used for the frequency and amplitude sweeps were 0.1–150 rad/s and 0.01–1000% (points per decade 5) at the constant strain and frequency of 1.0% and 1.0 rad/s, respectively. Flow properties of the test sample were determined over a shear rate range of 0.01–1000 s^−1^ (points for decade 5) with the maximum equilibration time of 60 s using Steady state sensing function in TRIOS software (TA Instruments, version 4.3).

### 2.6. Water Vapor Transmission Rate

The water vapor transmission rate (WVTR) of the films was measured according to the monograph of the European Pharmacopoeia [20]. To determine moisture permeability, 10 mL of water was added to cylindrical bottle with 13 mm diameter containing. Then, the chitosan films were cut into 15 mm diameter discs and then mounted onto the bottle followed by sealing with parafilm. The bottle was then kept in an oven at 45 °C for 24 h. The WVTR was calculated using the following Equation (1):(1)$$\mathrm{WVTR}=\frac{\left(w_{i}-w_{t}\right)}{A\times 24}\;g{\;m}^{-2}h^{-1}$$
where *A* is the area of the bottle mount (m^2^), and $w_{i}$ and $w_{t}$ are the weight of the bottle at time zero and the weight of the bottle after 24 h.

### 2.7. Bio Screening

#### 2.7.1. Cultures

The test strains *Escherichia coli* (ATCC 8739) and *Staphylococcus aureus* (ATCC 6538) were incubated at 37 °C for 24 h. The cultures were stored frozen at −80 °C in cryovials (Microbank, Pro-Lab Diagnostics, Richmond Hill, ON, Canada). Before each experiment, a frozen bead was recovered in Tryptone Soy Broth (Oxoid, Basingstoke, UK). Tryptone soya broth (TSB) was prepared by dissolving 15g TSB in 500 ml distilled water on a stirrer before adjustment of the pH to 7.1 ± 0.1 by use of HCl. The broth was sterilized by autoclaving at 121 °C for 15 min. The broth was cooled on the bench before storage in a cold room at 4 °C for a maximum of 30 days.

#### 2.7.2. Bioscreen Experiments

Each overnight-recovered suspension was inoculated in a 100 mL erlenmeyer flask containing 25 mL TSB and grown to stationary phase (growth conditions: 20 h/30 °C/150 rpm). Tenfold serial dilutions of the inoculum were prepared in TSB with five dilutions (10^−2^ to 10^−6^) followed by transferring each strain to 100 well microtiter plates (300 µL in each well). The microtiter plates were mounted in a Bioscreen C (Oy Growth Curves Ab Ltd., Helsinki, Finland) programmed to measure absorbance at 600 nm (abs 600 nm) at regular time intervals. Prior to each measurement, the plates were shaken for 10 s (default setting). The temperatures investigated were 30 °C. For each strain, a calibration curve was set up to relate turbidity measurements (abs 600 nm) to a dilution series of the strains.

### 2.8. Statistical Analysis

Average values and standard deviation were computed, and statistical analysis was performed using the Prism software package (version 3.02, GraphPad Software, San Diego, CA, USA). Two-way analysis of variance (ANOVA) was performed with the Bonferroni post-test to compare the significance of change in one factor with time. The error bars presented represent standard deviation with 𝑛 = 3.

## 3. Results and Discussion

### 3.1. FTIR Spectra Analysis

Infrared spectroscopy is an important tool for identifying interactions between chitosan and different acids and their associated functional groups. Figure 1 shows the transmission FTIR spectra of the chitosan solution prepared using acetic acid (2% *v*/*v*), a combination of acetic acid (1% *v*/*v*) and citric acid (1% *w*/*v*), and citric acid (2, 4, and 6% *w*/*v*). For all the samples, there is a strong, broad absorbance from 2400–3500 cm^−1^.

The absorbance band resulted from O-H stretching overlapping with N-H stretching vibrations. The main absorbance and band width for the different samples varies as the vibrational bands in this region are highly influenced by hydrogen bonding that is highly dependent on the functional groups involved. For example, the region around 3400 cm^−1^ can be affected by contributions from solvated water [24], which might explain the visible variations in the intensity combined with a slight shift in the local maximum (3421 cm^−1^ for CH-AA shifting to 357 cm^−1^). At the same time, the -OH contributions from carboxylic acid moieties occur between 2400–2800 cm^−1^ and this is clearly reflected in all the CH-CA spectra. In the spectral region from 1800 cm^−1^ and downwards, the CH-CA samples are dominated by a peak at 1725 cm^−1^ caused by C=O stretching in the carboxylic acids, which can be attributed to the protonated form of the carboxylic acid moieties in the CH-CA samples. This band is clearly increasing with the acid concentration of citric acid. The position of this band reveals a slight shift to lower wavenumbers (1718 cm^−1^) while the band decreases in intensity for the CH-CA/AA and it completely disappears for CH-AA, which is in accordance with observations made by Garcia et al [25]. Spectral bands between 1700–1500 cm^−1^ are often indicative of Amide I and Amide II vibrations. However, the asymmetric C=O stretch from carboxylate and deformation vibration of NH^3+^ groups also occur in this region [3], as well as amine deformation vibrations producing strong bands in the region between 1638 to 1575 cm^−1^ [26]. This renders the assignment of clear bands a challenging task for more complex molecules or mixtures of molecules. Lawrie et al. reported that chitosan with an 85% degree of deacetylation displays two vibrations at 1645 and 1584 cm^−1^ that are assigned to amide I and amide II vibrations [27]. Therefore, it could be proposed that the two vibration bands observed in the region of 1584 and 1645 cm^−1^ for CH-2% AA solution were due to the N-H bending vibrations overlapping the amide II vibration and the amide I vibration, respectively. All the samples display a clear band at around 1407 cm^−1^ due to the symmetric stretching vibration in COO- [24]. At the same time, the asymmetric stretch for the COO- groups appears only as a shoulder at 1622 cm^−1^ for CH-CA, while there is a strong band at 1558 cm^−1^ displayed for CH-AA. For CH-AA, a shoulder at 1637 cm^−1^ appears just under the 1558 cm^−1^ band and is likely a contribution from C=O in the amide bond of the acetylated part of chitosan (85% degree of acetylation), which disappears under the large peak from the carboxylic acid functional group for the CH-CA samples. The small band at 1516 cm^−1^ visible for CH-CA is indicative of the presence of NH^3+^ functional groups [24], indicative of the protonation of -NH_2_ moieties of chitosan. For CH-AA, this vibrational mode may be hidden under the much more pronounced band at 1558 cm^−1^. The acetic acid in CH-AA appears to be fully deprotonated as there is no band present between 1800-1700 cm^−1^. The bands found in the fingerprint region 1300–1000 cm^−1^ are mainly due to C-O-C and C-O stretching vibrations with a major contribution from the glycosidic entities present in the chitosan chains for all the samples. The large contribution for CH-CA at 1215 cm^−1^ is likely due to C-O found in the citric acid as they clearly increase with an increasing concentration of the acid. Lawrie et al. also reported that the C-N stretching vibrations appear in the region between 1190–920 cm^−1^ and overlap with the vibrations from the carbohydrate ring [27]. More specifically, the skeletal vibration of C-O stretching and the antisymmetric stretch of C-O-C and C-N appear at 1018 and 1151 cm^−1^, respectively [28]. The presence of characteristic peaks from citric acid proved that citric acid was not removed during the solvent-casting process. The presence of numerous absorption bands from citric acid and the bands resulting from the probable interaction between chitosan and citric acid caused an extensive overlapping between the chitosan and citric acid bands resulting in a very complex spectrum.

### 3.2. Mechanical Properties

Figure 2 shows the tensile strength (TS) and elongation at break (EB) of chitosan films prepared with different acidic solutions. The tensile strength and elongation of break of the films were greatly affected by the type of acid used. In general, the citric acid-containing films showed lower tensile strengths compared to the acetic acid films. A reduction in TS was also observed with an increasing citric acid content. On the contrary, the EB of the citric acid-containing films were significantly higher than the acetic acid films. The TS of chitosan films prepared with 2% *v*/*v* acetic acid (CH-2% AA) was 53.51 MPa, which reduced to 43.16 MPa as 2% *v*/*v* acetic acid solution was replaced with a combination of 1% *v*/*v* acetic acid and 1% w/v citric acid (CH-1% AA-1% CA). The TS of the 2% *w*/*v* (CH-2% CA) citric acid films were 13.44 MPa, which reduced to 10MPa and 3MPa as the citric acid concentration was increased to 4% *w*/*v* (CH-4% CA) and 6% *w*/*v* (CH-6% CA), respectively. The TS of the CH-2% CA films were 70% lower than the TS of the CH-2% AA films. The EB of the CH-2% AA films were 5% which increased to 31% for CH-1% AA-1%CA films. A further increase in EB to 81% was observed as 2% acetic acid was completely replaced by 2% citric acid. The highest EB (220%) was observed for the CH-4% CA and CH-6% CA films. The lower TS of the CH-CA films were mainly due to the presence of residual acid in the films [20]. The residual acid may reduce the interaction between the polymer chains due to the poor packing resulting in a reduced TS. On the other hand, this reduced interaction between polymer chains allows them to slide over each other, which can significantly increase the EB [28]. Therefore, the absence of this residual acid in the CH-AA films enabled better packing within the chitosan polymer chain resulting in stronger films. Uranga et al. studied the effect of the addition of citric acid (10 and 20 wt%) on the mechanical properties of gelatin/chitosan composite films. They also reported an increase in the elongation at break values and a decrease in the tensile strength of the composite films with an increasing citric acid content due to the plasticizing effect of citric acid [29].

Moreover, it is important to note that the pH of the citric acid samples was lower than the acetic acid samples, which reduced with an increasing citric acid content (Table 1). Low pH can lead to a decrease in the acetylation degree and partial depolymerization due to the hydrolysis of the chitosan chain resulting in a reduced molecular weight with a weak hydrogen bond network and reduced mechanical properties [30,31].

### 3.3. Rheological Properties

The amplitude sweep curves and the corresponding strain values extracted from the curves for the chitosan samples prepared using acetic acid, a combination of acetic acid and citric acid, and different concentrations of citric acid are presented in Figure 3 and Table 2, respectively. The samples prepared with citric acid showed shorter LVE regions that reduced with an increasing citric acid concentration as compared to the acetic acid samples. The linear viscoelastic (LVE) region for the chitosan films prepared using 2% *v*/*v* acetic acid was located at a strain value of 100%, which reduced to 63% as 2% *v*/*v* acetic acid solution was replaced with a combination of 1% *v*/*v* acetic acid and 1% *w*/*v* citric acid. The 2% *w*/*v* and 4% *w*/*v* citric acid-containing samples showed the same LVE region with a strain value of 39%. However, a further increase in the citric acid concentration up to 6% *w*/*v* citric acid reduced the LVE to a strain value of 25%. The strain values of the samples correlated well with the tensile strength data, confirming the reduced interaction between the polymer chains due to the presence of residual acid in the citric acid samples.

Figure 4 represents the relationship between the storage modulus (G′) and loss modulus (G″) vs the angular frequency from the oscillatory frequency sweep of the chitosan solution prepared with 2% *v*/*v* acetic acid, a combination of 1% *v*/*v* acetic acid and 1% w/v citric acid, and 2, 4, and 6% *w*/*v* citric acid. In addition, Figure 5 represents the time sweep profiles of the storage modulus (G′) and loss modulus (G″) of the chitosan solution prepared using different acidic solutions. For all the samples, in the beginning, the G″ was higher than G′, which is consistent with the other findings in the literature [9]. It has been reported that the initial higher value of G″ as compared to G′ is due to the fact that the samples are still in the liquid state; therefore, the viscous properties are more dominant [9]. As the time advanced, both G′ and G″ increased and due to the formation of the cross-linked chitosan structure the solution turned into a gel-like state. However, the rate of increase in G′ with time was greater than the G″, which shows that the elastic properties became more dominant, resulting in a G′ and G″ crossover. The time required to achieve this cross over indicates the gelation time. As seen from Figure 4, the gelation times for CH 2% AA, CH 1% AA-1% CA and CH 2% CA, and CH 4% CA and CH 6% CA were around 230s and 240s. Therefore, it could be concluded that different acid concentrations did not have a large effect on the gelation time.

The choice of acid not only plays a key role in the dissolution of chitosan but also affects the solution properties such as viscosity [28]. Figure 6 represents the effect of acetic acid, a combination of acetic and citric acid, and different concentrations of citric acid on the solution viscosity of chitosan solution. Chitosan solution prepared using a solution of 1% acetic acid and 1% citric acid (CH 1% AA-1% CA) showed the lowest solution viscosity. The solution viscosity of the CH 2% AA was higher than CH 1% AA-1% CA. However, only CA-containing samples showed a higher viscosity as compared to the AA-containing solution, which increased with an increasing CA content. Moreover, a more marked shear-thinning effect was observed with only the CA-containing samples. The addition of CA enables more interchain interactions, resulting in more pronounced shear-thinning effects [20]. Viscosity measurements can indicate the hydrodynamic behavior of macromolecules in solutions and their interactions with solvent [32]. Chen et al. studied the changes in the rheological properties of the chitosan solutions with pH change and reported that between pH 2.0 and 5.0 apparent viscosities increased with the decrease in pH for chitosan in organic acid solutions [33]. It has been reported that due to the deprotonation of amino groups and the loss of charge the chitosan macromolecules, the relative viscosity can significantly decrease with an increasing pH [32]. As reported in Table 1, the pH of the chitosan solution decreased with an increasing CA content and ranged between 2.80 to 2.45. Therefore, it can be concluded that the decreasing pH with an increasing CA content might have caused the deprotonation of the amino groups and the loss of charge in the chitosan molecules resulting in an increased solution viscosity.

### 3.4. Water Vapour Transmission Rate

Figure 7 shows the effects of different acid concentrations on the water vapor transmission rate (WVTR) of the chitosan films. No statistical difference in the WVTR was observed between the chitosan films prepared with 2% *v*/*v* acetic acid and a combination of 1% *v*/*v* acetic acid and 1% *w*/*v* citric acid. However, the WVTR rate of the films prepared with only citric acid was lower than the acetic acid-containing films, which shows their superior barrier properties. The solvent used to prepare chitosan solution plays an important role in deciding the WVTR of the films. The hydrophilic nature of chitosan leads to a higher interaction with water molecules that in turn leads to a high WVTR [34]. The reduction in the WVTR rate could be due to the decreased availability of hydrophilic hydroxyl groups and the generated hydrophobic ester groups between citric acid and the polysaccharides, resulting in a denser structure and thus an improved WVTR.

### 3.5. Bioscreen Results

Growth curves were produced in triplicate for cell concentrations of 10^−2^ to 10^−6^ for both *S. aureus* and *E. coli* in nutrient broth using the Bioscreen. The mean of the three sets of data was calculated and the resulting growth curves were plotted (Figure 8). It can be seen that the higher the initial cell number, the faster the time to detection was reached. The detection time is determined from a point where there is a rapid increase in OD and can be related to the number of cells present. TTD is the time to detection, i.e., the time it takes for the abs 600 nm to reach 0.2. Having determined the detection time for each cell number, these were used to construct a calibration graph (Figure 9). These results show that there is a very good correlation between the OD data and cell numbers, with r^2^ = 1.00 for *E. coli* and 0.99 for *S. aureus* (Figure 9).

#### Testing with Chitosan Solution

Figure 10 shows the optical density (OD) of TSB broth inoculated with *E. coli* and *S. aureus* during its incubation time at 30 °C. A faster rate of increase in the OD and higher OD values were observed in the control culture compared with those with added chitosan during their inscubation times. The TDT for *E. coli* control was 250 min and for the S. aureus control was 564 min. All concentrations of chitosan (CH 2% AA, CH 2% CA, CH 6%CA, CH 1% AA-1% CA, and CH4% CA) suppressed the growth of the two test strains, and none of the variants for an abs 600 nm at 0.2.

## 4. Conclusions

In the present study, an attempt was made to establish a one-step process to dissolve and crosslink chitosan solution using citric acid at different concentrations. The effects of such a one-step process on the mechanical, rheological, barrier-related, and antimicrobial properties of chitosan films and solution were evaluated. A combination of acetic and citric acid and three different concentrations of citric acid were also used to simultaneously dissolve and crosslink chitosan solution. A decrease in the tensile strength and a remarkable increase in the elongation at break was observed with the increased amount of citric acid in the films. Such changes in the mechanical properties were due to the presence of excess citric acid in the films, which was also confirmed via the FTIR analysis. The amplitude sweep data from the rheological experiment demonstrated that the samples prepared with citric acid showed shorter LVE regions that reduced with an increasing citric acid concentration as compared to the acetic acid samples. The amplitude sweep data correlated well with the tensile strength data, confirming the reduced interaction between polymer chains due to the presence of residual acid in the citric acid samples. The solution viscosities of the citric acid-containing samples were higher than the acetic acid samples that further increased with an increasing citric acid concentration. Such an increase in viscosity was due to the deprotonation of the amino groups and the loss of charge in the chitosan molecules resulted from the decreasing pH with an increasing citric acid concentration. The water vapor transmission rate of the citric acid samples was lower than the acetic acid samples. All the concentrations of chitosan (CH 2% AA, CH 2% CA, CH 6%CA, CH 1% AA-1% CA, and CH4% CA) suppressed the growth of the two test strains, and none of the variants reached an abs 600 nm at 0.2.

## Figures and Tables

**Figure 1 molecules-27-05118-f001:**
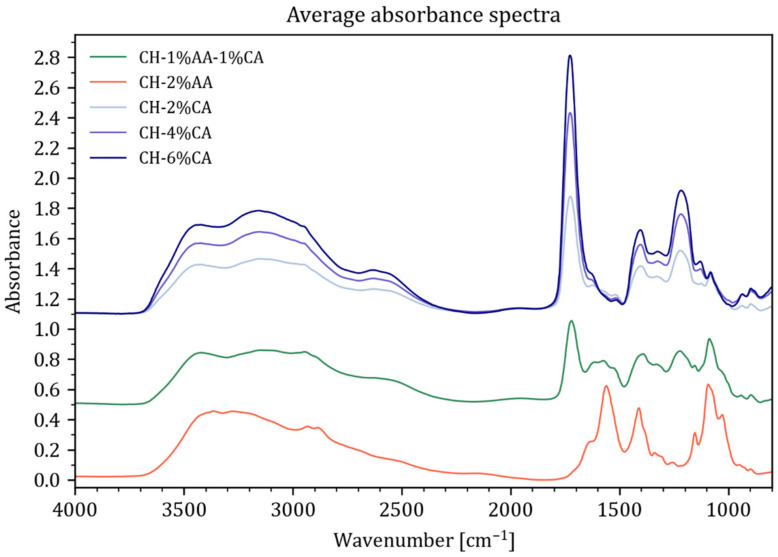
Dry-film transmission FTIR spectra of the chitosan films prepared using acetic acid (2% *v*/*v*), a combination of acetic acid (1% *v*/*v*) and citric acid (1% *w*/*v*), and citric acid (2, 4, and 6% *w*/*v*).

**Figure 2 molecules-27-05118-f002:**
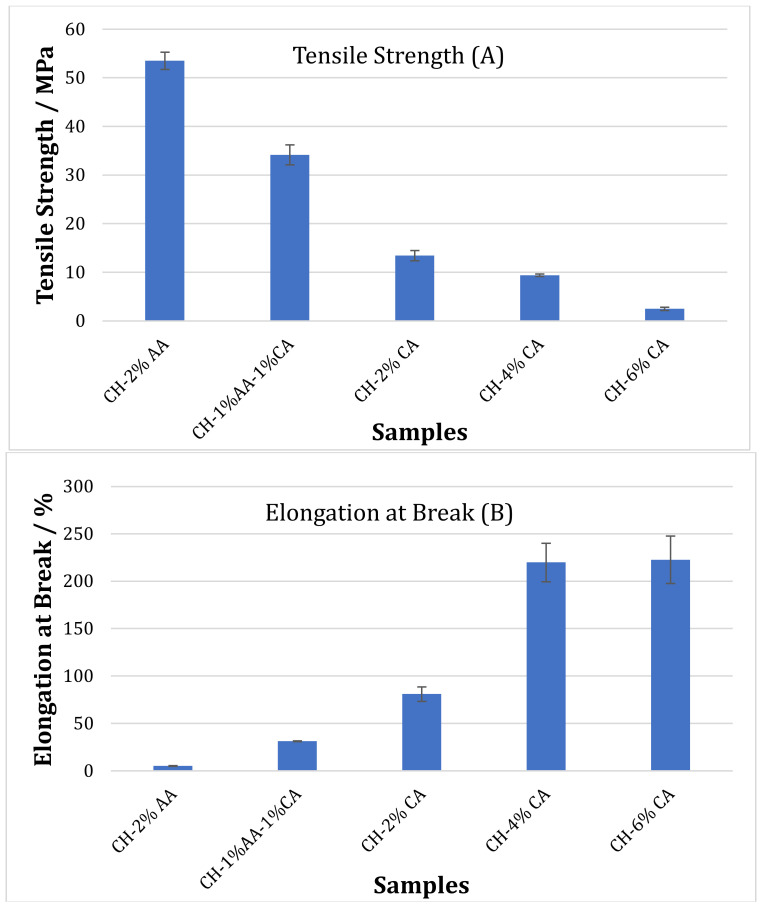
Tensile Strength (**A**) and Tensile Modulus (**B**) of the chitosan films prepared using acetic acid (2% *v*/*v*) and citric acid (2, 4 and 6% *w*/*v*).

**Figure 3 molecules-27-05118-f003:**
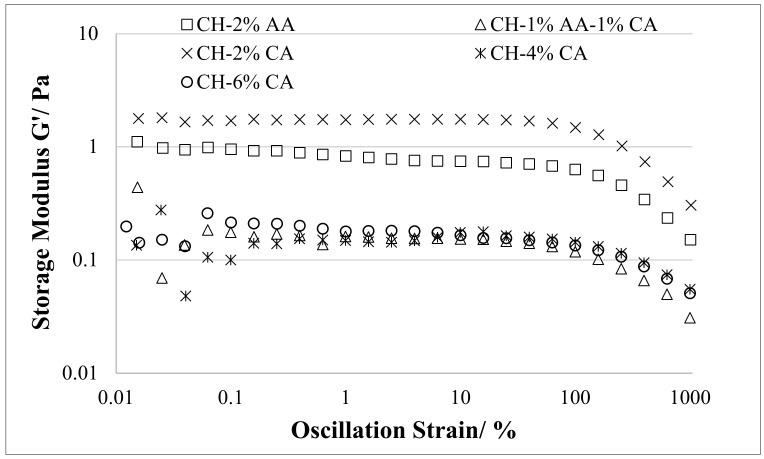
Oscillatory amplitude sweeps of chitosan solutions prepared using acetic acid (2% *v*/*v*) and citric acid (2, 4 and 6% *w*/*v*).

**Figure 4 molecules-27-05118-f004:**
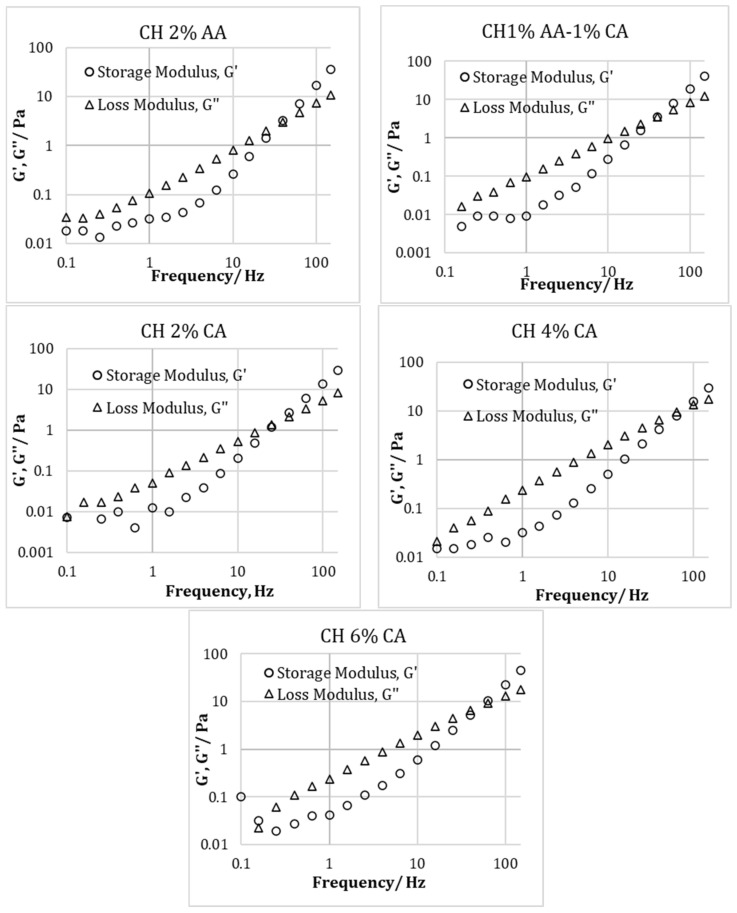
Relationship between Storage modulus (G′) and loss modulus (G″) vs. angular frequency from oscillatory frequency sweep of chitosan solution prepared using acetic acid (2% *v*/*v*) and citric acid (2, 4, and 6% *w*/*v*).

**Figure 5 molecules-27-05118-f005:**
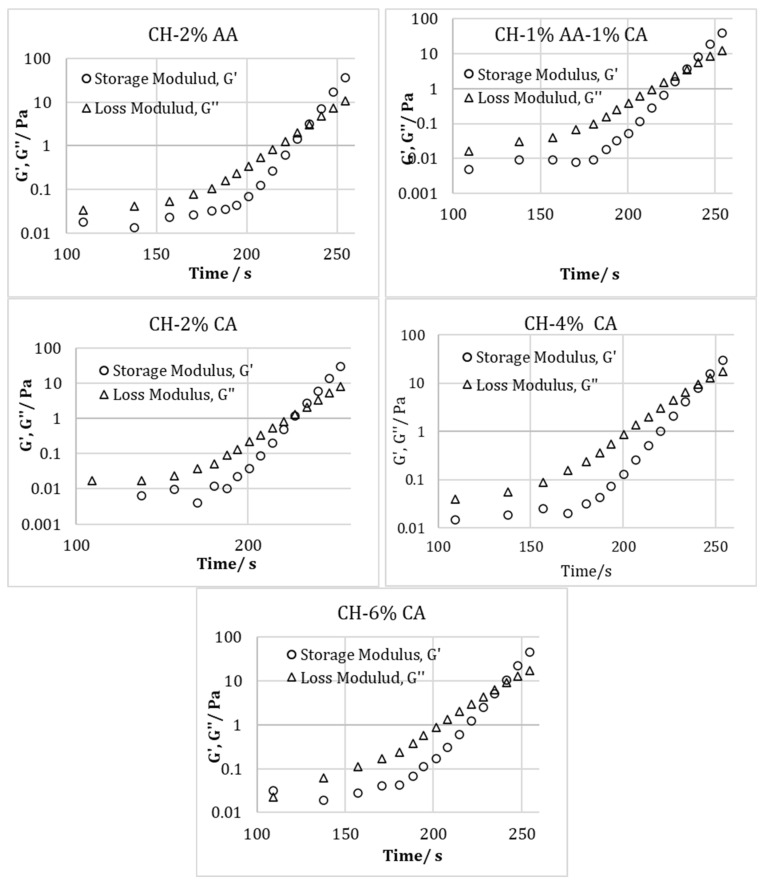
Relationship between Storage modulus (G′) and loss modulus (G″) vs time from oscillatory frequency sweep of chitosan solution prepared using acetic acid (2% *v*/*v*) and citric acid (2, 4, and 6% *w*/*v*).

**Figure 6 molecules-27-05118-f006:**
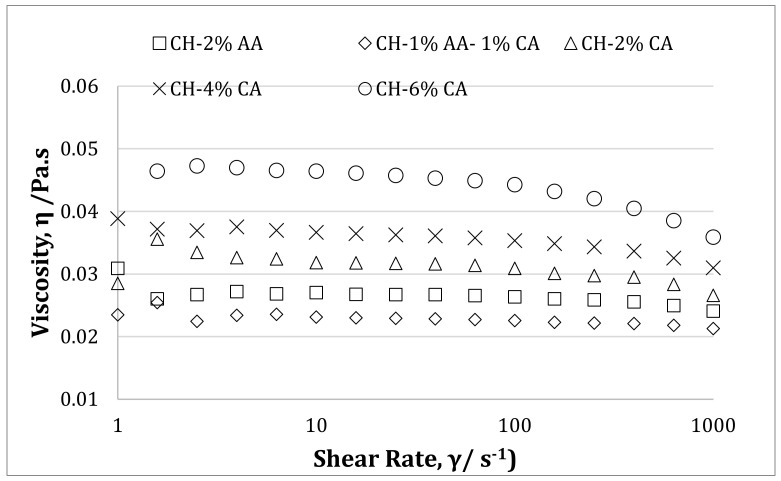
Shear flow curves of chitosan solution prepared with acetic acid (2% *v*/*v*) and citric acid (2, 4, and 6% *w*/*v*).

**Figure 7 molecules-27-05118-f007:**
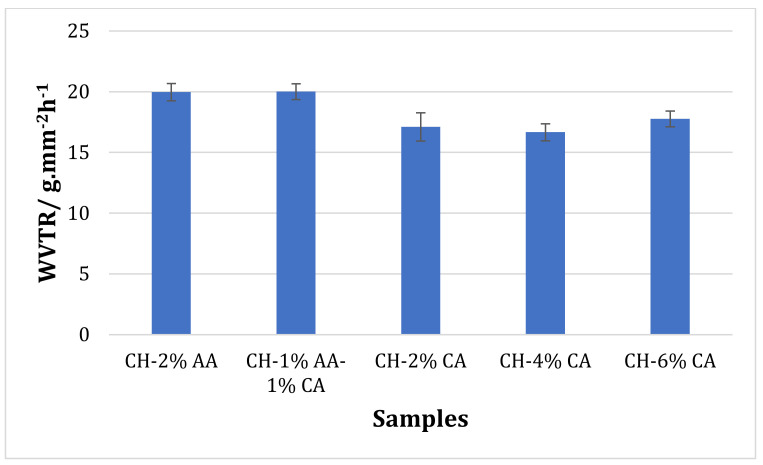
Water vapor transmission rate (WVTR) of chitosan films prepared using acetic acid (2% *v*/*v*) and citric acid (2, 4, and 6% *w*/*v*).

**Figure 8 molecules-27-05118-f008:**
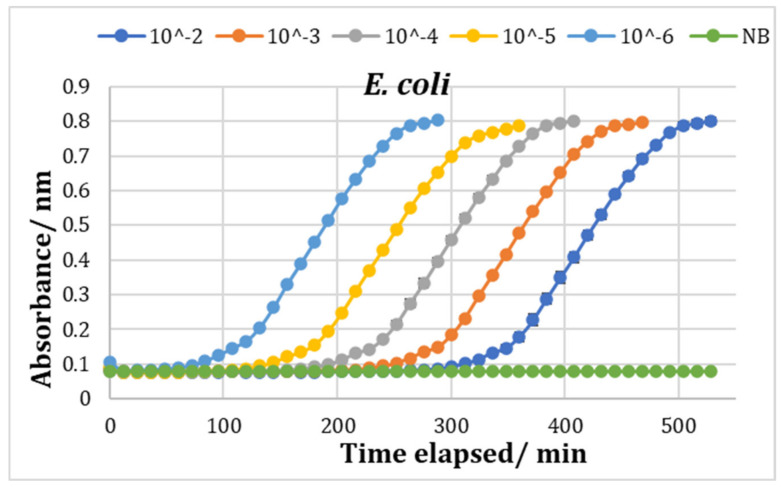
Growth curves of different concentrations of *E. coli* and *S. aureus*. All data points are averages of three parallel samples, with error bars representing standard deviation. The dotted black line represents the detection limit, Abs600 = 0.2.

**Figure 9 molecules-27-05118-f009:**
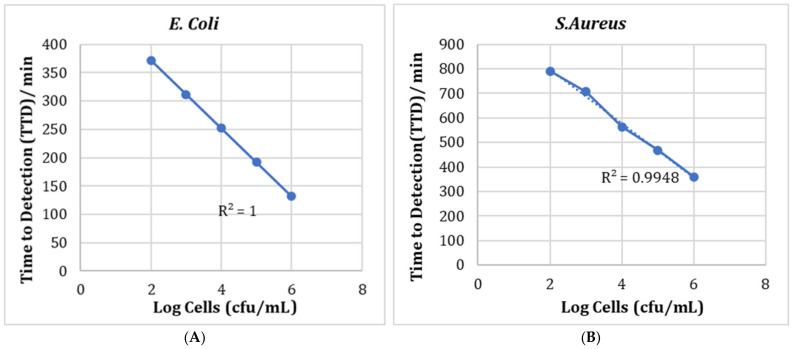
Calibration curves for *E. coli* (**A**) and *S. aureus* (**B**) with detection limit at Abs600 = 0.2.

**Figure 10 molecules-27-05118-f010:**
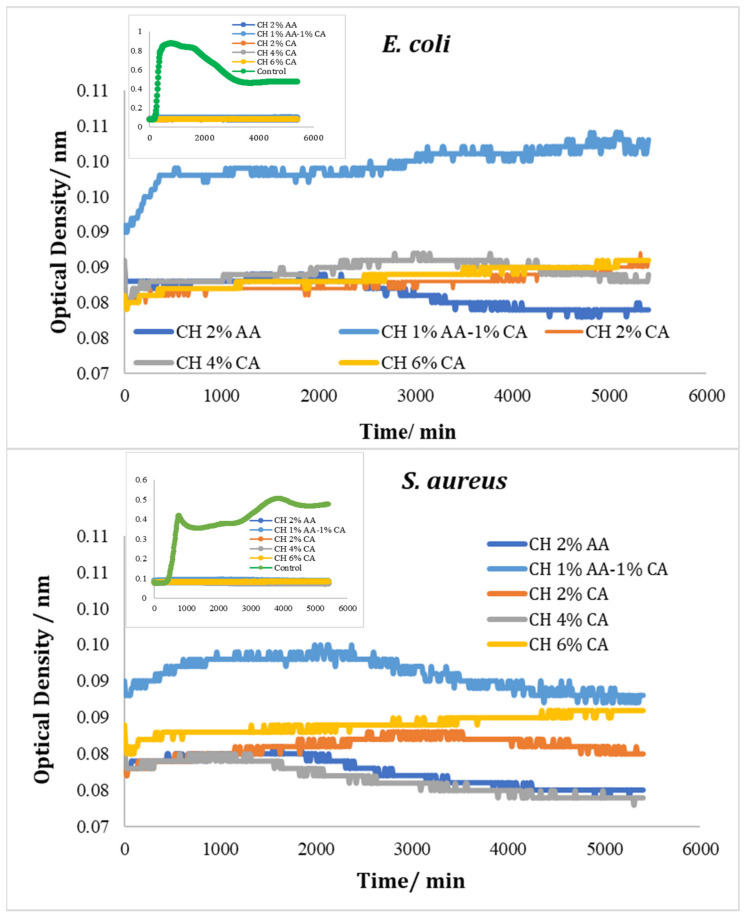
Growth curves of *E. coli* and *S. aureus* with different added chitosan concentrations in AA and CA solutions. All data points are averages of three parallel samples, with error bars representing standard deviation. The small inserted figures show the growth curves of the chitosan solution against the control, showing Abs (600 nm) from 0.07 to 0.11 nm.

**Table 1 molecules-27-05118-t001:** The pH of the chitosan solution along with control.

Sample	Sample Code	pH
2% Acetic acid	2% AA	2.63
Chitosan with 2% Acetic acid	CH-2% AA	2.83
2% Citric acid	2% CA	2.47
Chitosan with 2% Citric acid	CH-2%CA	2.76
4% Citric acid	4% CA	2.35
Chitosan with 4% Citric acid	CH-4%CA	2.46
6% Citric acid	6% CA	2.31
Chitosan with 6% Citric acid	CH-6%CA	2.45
1% Acetic acid + 1% Citric acid	1% AA+ 1% CA	2.42
Chitosan with 1% Acetic acid + 1% Citric acid	CH-1%AA-1%CA	2.68

**Table 2 molecules-27-05118-t002:** Limit strain of the LVE % of chitosan solution prepared using acetic acid (2% *v*/*v*) and citric acid (2, 4 and 6% *w*/*v*).

Samples	Strain %
CH-2% AA	100
CH-1% AA-1% CA	63
CH-2% CA	39
CH-4% CA	39
CH-6% CA	25

## Data Availability

Not applicable.

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
