# Peer review of "Effect of Citric Acid Cross Linking on the Mechanical, Rheological and Barrier Properties of Chitosan"

_molecules, 2022, doi:10.3390/molecules27165118_

Round 1

Reviewer 1 Report

In this work, the authors investigated the effect of citric acid on the crosslinking of chitosan and its mechanical, rheological, and barrier properties. The article is written in good language with clear formulations, and a large amount of data ennobles this work. I would highlight the following points to be corrected:

1. Remove color highlighting throughout the text.

2. Why did the authors choose citric acid and acetic acid in particular?

3. All equations must be numbered.

4. It is desirable to show the reaction equation and the experimental scheme. This will clarify some points.

5. It is desirable in the introduction (and in other parts) to indicate the scope and analogues of the substance obtained.

6. In addition, in the introduction, you can refer to: 10.1016/j.molstruc.2021.131083 and others.

7. When describing the optimization / mathematical model of the experiment, you can refer to 10.1007/s13399-021-01895-y and others.

8. Conclusions can be made more concise.

9. Why did the authors choose such a temperature and duration for the experiment?

10. In general, I would recommend that the authors add more references to the literature. And also compare the obtained data with those known from the literature. For example, comparison with starch citrates and other polysaccharides.

Author Response

In this work, the authors investigated the effect of citric acid on the crosslinking of chitosan and its mechanical, rheological, and barrier properties. The article is written in good language with clear formulations, and a large amount of data ennobles this work. I would highlight the following points to be corrected:

  1. Remove color highlighting throughout the text.

Response: The highlights have been removed.

  1. Why did the authors choose citric acid and acetic acid in particular?

Response: Acetic acid is mostly used in dissolving chitosan and citric acid can be used as a cross linking agent. The author’s wanted to evaluate how the use of only citric acid instead of acetic acid and a combination of citric acid & acetic acid can affect the dissolution process and the functional properties of the final film.

  1. All equations must be numbered.

Response: Modified as suggested.

  1. It is desirable to show the reaction equation and the experimental scheme. This will clarify some points.

Response: The author’s wanted to add the reaction equation to the manuscript as well. However, it was not included considering the following facts:

It was expected that when a combination of acetic acid and citric acid was used, acetic acid will help to create the acidic media which will eventually dissolve the chitosan and the citric will act as a crosslinking agent and hence increase the functional properties However, the tensile strength of the acetic acid & citric acid containing samples were lower than the only acetic acid containing samples due to the presence of excess citric acid in the samples (also evident from the FTIR). Under such circumstances, if we show a cross linking reaction between chitosan and citric acid, it might create some confusions between the readers. That’s why the reaction equation was not added and was rather explained in the text.

  1. It is desirable in the introduction (and in other parts) to indicate the scope and analogues of the substance obtained.

Response: The Introduction section has been modified as suggested.

  1. In addition, in the introduction, you can refer to and others.

Response: Added as suggested.

  1. When describing the optimization / mathematical model of the experiment, you can refer to 10.1007/s13399-021-01895-y and others.

Response: The suggested paper is on the optimization of guar gum galactomannan sulfation process with sulfamic acid, Unfortunately, the authors could not connect the current work with the suggested paper. That’s why we could not add the paper.

  1. Conclusions can be made more concise.

Response: The conclusion section has been concise.

  1. Why did the authors choose such a temperature and duration for the experiment?

Response: All the experiments were deigned following some standard or some previously published experiment.

  1. In this work, the authors investigated the effect of citric acid on the crosslinking of chitosan and its mechanical, rheological, and barrier properties. The article is written in good language with clear formulations, and a large amount of data ennobles this work. I would highlight the following points to be corrected:

    1. Remove color highlighting throughout the text.

    Response: The highlights have been removed.

    1. Why did the authors choose citric acid and acetic acid in particular?

    Response: Acetic acid is mostly used in dissolving chitosan and citric acid can be used as a cross linking agent. The author’s wanted to evaluate how the use of only citric acid instead of acetic acid and a combination of citric acid & acetic acid can affect the dissolution process and the functional properties of the final film.

    1. All equations must be numbered.

    Response: Modified as suggested.

    1. It is desirable to show the reaction equation and the experimental scheme. This will clarify some points.

    Response: The author’s wanted to add the reaction equation to the manuscript as well. However, it was not included considering the following facts:

    It was expected that when a combination of acetic acid and citric acid was used, acetic acid will help to create the acidic media which will eventually dissolve the chitosan and the citric will act as a crosslinking agent and hence increase the functional properties However, the tensile strength of the acetic acid & citric acid containing samples were lower than the only acetic acid containing samples due to the presence of excess citric acid in the samples (also evident from the FTIR). Under such circumstances, if we show a cross linking reaction between chitosan and citric acid, it might create some confusions between the readers. That’s why the reaction equation was not added and was rather explained in the text.

    1. It is desirable in the introduction (and in other parts) to indicate the scope and analogues of the substance obtained.

    Response: The Introduction section has been modified as suggested.

    1. In addition, in the introduction, you can refer to and others.

    Response: Added as suggested.

    1. When describing the optimization / mathematical model of the experiment, you can refer to 10.1007/s13399-021-01895-y and others.

    Response: The suggested paper is on the optimization of guar gum galactomannan sulfation process with sulfamic acid, Unfortunately, the authors could not connect the current work with the suggested paper. That’s why we could not add the paper.

    1. Conclusions can be made more concise.

    Response: The conclusion section has been concise.

    1. Why did the authors choose such a temperature and duration for the experiment?

    Response: All the experiments were deigned following some standard or some previously published experiment.

    1. In general, I would recommend that the authors add more references to the literature. And also compare the obtained data with those known from the literature. For example, comparison with starch citrates and other polysaccharides.

    Response: More references has been added to the article.

    In general, I would recommend that the authors add more references to the literature. And also compare the obtained data with those known from the literature. For example, comparison with starch citrates and other polysaccharides.

Response: More references has been added to the article.

Reviewer 2 Report

Dear Authors,

The work has been revised and supplemented according to the reviewer's comments. In its present form, it can be accepted.

Best regards,

Reviewer

Author Response

Thank you very much for suggesting our article for publication.

Reviewer 3 Report

Unfortunately, the authors did not address my comments regarding the novelty of the study (#1) and the evidence of cross-linking (#2). Therefore, I do not recommend this paper for publication.

1.       The introduction contains much general textbook information, but there is no critical review of the literature on the properties of the systems studied. Although the chitosan-acetic acid and chitosan- citric acid systems have been studied for decades. In this regard, it is essential to describe the novelty of the presented work.

2.       The authors declare that the main objective of the work is the study of the effect of citric acid cross-linking on the functional properties of chitosan. However, the authors did not prove/determine the presence of cross-linking, the type of cross-linking (ionic or covalent) and the degree of cross-linking. From the presented work, we do not even know whether soluble or insoluble films were prepared. If the films are insoluble, what is the degree of swelling? Under what conditions can di-ester linkages with the hydroxyl groups of the polysaccharide be formed (line 70)?

Author Response

Unfortunately, the authors did not address my comments regarding the novelty of the study (#1) and the evidence of cross-linking (#2). Therefore, I do not recommend this paper for publication.

  1. The introduction contains much general textbook information, but there is no critical review of the literature on the properties of the systems studied. Although the chitosan-acetic acid and chitosan- citric acid systems have been studied for decades. In this regard, it is essential to describe the novelty of the presented work.

Response: The authors agree that there are similar works which have used citric acid as a cross linker. However, in this study we aimed to establish a one step process of dissolving and crosslinking chitosan using citric acid. So citric acid was used not only as a cross linking agent but also used to create the acidic media required to dissolve chitosan. Moreover, a combination of acetic & citric acid and different concentration of citric acid solution was also prepared to compare the effect of different acidic media on the functional properties of chitosan films.

  1. The authors declare that the main objective of the work is the study of the effect of citric acid cross-linking on the functional properties of chitosan. However, the authors did not prove/determine the presence of cross-linking, the type of cross-linking (ionic or covalent) and the degree of cross-linking. From the presented work, we do not even know whether soluble or insoluble films were prepared. If the films are insoluble, what is the degree of swelling? Under what conditions can di-ester linkages with the hydroxyl groups of the polysaccharide be formed (line 70)?

Response: Based on reviewer’s suggestion, authors did conduct some nin hydrin test to find out the degree of cross linking. However, the results were somehow inconclusive, hence not included in the article.

Round 2

Reviewer 1 Report

Accepted.

Reviewer 3 Report

Accept

This manuscript is a resubmission of an earlier submission. The following is a list of the peer review reports and author responses from that submission.

Round 1

Reviewer 1 Report

The paper ‘Effect of citric acid cross linking on the functinal properties of chitosan’ can be accepted after major revision. Please revise and work on following comments:

  1. ‘The experiment was conducted at 22oC temperature’:

Correct the symbol

  1. Why WVTR equation is multiplied with 10^6 is not clear.
  2. ‘10-fold serial dilutions of the inoculum were prepared in TSB with 5 dilutions (10- 2 to 10-6 ) of each strain were transferred to 100 well microtiter plates’ sentence construction is totally pooor.
  3. Plagiarism of the manuscript should be checked.
  4. Why LVE region will decrease in case of CA? Residuals are also present in other acids such as AA etc
  5. There are many works similar which have used citric acid as crosslinker. So what is the novelty of this work?
  6. Can go through some references which have worked on chitosan, crosslinking etc: https://doi.org/10.1016/j.nanoso.2020.100425; https://doi.org/10.1016/j.ijbiomac.2019.02.117; https://doi.org/10.1016/j.ijbiomac.2020.08.060
  7. The conclusion can be impproved.

Reviewer 2 Report

Dear Authors,

The design of the experiment is adequate to test the hypothesis. The title, however, is too general and only after reading the contents of the manuscript does the reader become aware of the scope of the topic. I suggest making the title more specific so that it relates to the subject matter presented in the paper.

The results presented are clear and understandable, and the data are interpreted appropriately and consistently throughout the manuscript.

Of the 25 citations, only 7 of them are from the last five years. It is recommended that the literature be updated. The discussion also lacks comparisons with available literature data to demonstrate the novelty of the work. As far as I know, there have already been studies in which chitosan was dissolved in citric acid and also in acetic acid. Therefore, please make comparisons of your findings with data from the last 5 years.

The conclusions have been correctly formulated and presented. 

The abstract does not fully reflect the manuscript. It needs to be revised.

Best regards,

Reviewer

Reviewer 3 Report

This study was aimed to establish a one step process to dissolve and crosslink chitosan solution using a mixture of acetic and citric acids. This topic has already been dealt with in numerous works over the past decades, so the novelty of the work is low. I do not recommend the publication of this paper for the following main reasons:

  1. The introduction contains much general textbook information, but there is no critical review of the literature on the properties of the systems studied. Although the chitosan-acetic acid and chitosan- citric acid systems have been studied for decades. In this regard, it is essential to describe the novelty of the presented work.
  2. The authors declare that the main objective of the work is the study of the effect of citric acid cross-linking on the functional properties of chitosan. However, the authors did not prove/determine the presence of cross-linking, the type of cross-linking (ionic or covalent) and the degree of cross-linking. From the presented work, we do not even know whether soluble or insoluble films were prepared. If the films are insoluble, what is the degree of swelling? Under what conditions can di-ester linkages with the hydroxyl groups of the polysaccharide be formed (line 70)?
  3. In my opinion, the culture tests were conducted methodologically inaccurately. No control solutions of acetic and citric acids (they are known to inhibit bacterial growth). Also, how do antibacterial experiments on chitosan solutions relate to the properties of the films?
  4. The paper also contains a significant number of errors, inaccuracies, and weird expressions. E.g. (not an exhaustive list):
    • Lines 34-36: Chitin (not chitosan) is the second most abundant biopolymer.
    • Line 37: Incomplete sentence, add “glycosidic bonds”.
    • Fig 7: The gram unit symbol is g.
    • Figs 8 and 10: First, the ordinate axis should be the optical density, not the absorbance. Second, either of these values are dimensionless and in no case can be expressed in nm.
    • Line 402: “none of variants reached an abs 600 nm at 0.2.” What does it mean and why is it needed in the Conclusion?